

# Preserving and vouchering butterflies and moths for large-scale museum-based molecular research

Soowon Cho[1,2], Samantha W. Epstein[3], Kim Mitter[2], Chris A. Hamilton[3], David Plotkin[3], Charles Mitter[2] and Akito Y. Kawahara[3]

[1] Department of Plant Medicine, Chungbuk National University, Cheongju, South Korea
[2] Department of Entomology, University of Maryland, College Park, MD, United States
[3] Florida Museum of Natural History, University of Florida, Gainesville, FL, United States

## ABSTRACT

Butterflies and moths (Lepidoptera) comprise significant portions of the world's natural history collections, but a standardized tissue preservation protocol for molecular research is largely lacking. Lepidoptera have traditionally been spread on mounting boards to display wing patterns and colors, which are often important for species identification. Many molecular phylogenetic studies have used legs from pinned specimens as the primary source for DNA in order to preserve a morphological voucher, but the amount of available tissue is often limited. Preserving an entire specimen in a cryogenic freezer is ideal for DNA preservation, but without an easily accessible voucher it can make specimen identification, verification, and morphological work difficult. Here we present a procedure that creates accessible and easily visualized "wing vouchers" of individual Lepidoptera specimens, and preserves the remainder of the insect in a cryogenic freezer for molecular research. Wings are preserved in protective holders so that both dorsal and ventral patterns and colors can be easily viewed without further damage. Our wing vouchering system has been implemented at the University of Maryland (AToL Lep Collection) and the University of Florida (Florida Museum of Natural History, McGuire Center of Lepidoptera and Biodiversity), which are among two of the largest Lepidoptera molecular collections in the world.

## INTRODUCTION

With over 157,000 described species, Lepidoptera (butterflies and moths) is one of the most diverse insect orders (*Van Nieukerken et al.*, *2011*). Lepidoptera are frequently collected for their beauty; consequently, they comprise significant portions of the world's natural history collections. Historically, butterflies and moths have been preserved as dried, pinned specimens with their wings spread, allowing for aesthetically pleasing displays and access to genitalia, the dissection of which has been a standard for Lepidoptera taxonomy for centuries (*Knolke et al.*, *2005*).

Within the last several decades, there has been a growing need for researchers to obtain molecular data from butterfly and moth specimens. Many of these researchers remove legs from specimens and extract DNA from them (e.g., *Hebert et al.*, *2004*; *Knolke et al.*,

Corresponding authors
Soowon Cho, chosoowon@gmail.com
Akito Y. Kawahara,
kawahara@flmnh.ufl.edu

*2005*; *Zimmermann, Wahlberg & Descimon*, *2000*), so that tissues can be obtained without compromising diagnostic morphological structures. However, dried and pinned specimens are often low in DNA yield, and many pinned Lepidoptera specimens have been rehydrated, either with direct injection of water or with a relaxing chamber, which can result in degraded DNA. In order to obtain sufficient quantities of DNA, destructive sampling approaches have been proposed for museum specimens, but such approaches can often result in little or no morphological voucher (*Wandeler, Hoeck & Keller*, *2007*).

At a time when large amounts of high-quality, purified nucleic acids are needed for the increasing number of molecular studies, museums are uniquely positioned to serve as an important resource for the preservation of molecular-grade tissues. A challenge to many molecular tissue preservation methods is that these approaches damage key structures needed for identification of the organism. *Brower* (*1994*), *Brower* (*1996*) and *Brower* (*2000*) described a procedure for Lepidoptera tissue preservation in which the head and thorax are preserved in ≥95% ethanol and stored in an ultra cold freezer. He suggested that the wings, abdomen, antennae, and palpi are glued to a 1 × 3 cm card, and designated as morphological vouchers to enable examination of diagnostic characters. While Brower's approach can yield potentially large amounts of nucleic acids from the head and thorax, appendages glued to a card can only be viewed from a single perspective (i.e., only one wing surface can be examined) and the amount of sample preparation time can be long, making it difficult to implement for large collections that wish to store many tissues.

Here we present a procedure for storing Lepidoptera tissues that has been successfully implemented at the University of Maryland (UMD) AToL Lep Collection as part of the Lepidoptera Tree of Life (LepTree) project (e.g., *Bazinet et al.*, *2013*; *Cho et al.*, *2011*; *Regier et al.*, *2009*; *Regier et al.*, *2013*; *Regier et al.*, *2015*), and at the University of Florida's McGuire Center for Lepidoptera and Biodiversity (MGCL), Florida Museum of Natural History (FLMNH), one of the largest Lepidoptera collections in the United States (*Kawahara et al.*, *2012*). Wings can be quickly accessioned and saved in clear plastic coin holders or laminated cards, and the remainder of the specimen is preserved as a tissue sample in a cryogenic freezer and stored for future use. This approach enables easy examination and specimen digitization using a flatbed scanner of both dorsal and ventral wing surfaces, preservation of diagnostic body parts, and long-term storage of high-quality, molecular-grade tissues for nearly all body regions.

## METHODS

### Step 1: specimen sampling

Lepidoptera are collected in many different ways. Butterflies are typically caught with aerial nets, and nocturnal moths are often lured to a single site using a visual or olfactory stimulus, such as ultraviolet light or fermenting sugar, and trapped in a jar containing a killing agent (e.g., *Brehm & Axmacher*, *2006*; *Utrio*, *1983*). Although common Lepidoptera killing agents (i.e., cyanide, ethyl acetate) have been shown not to impact recovery of mtDNA COI barcodes (*Willows-Munro & Schoeman*, *2015*), there is still some debate as to whether they can have deleterious effects on genomic DNA (*Dean & Ballard*, *2001*; *Dillon, Austin*

*& Bartowsky*, *1996*). If RNA is required (e.g., for transcriptomics), the best practice is to collect the specimen without using a killing agent and flash-freeze it in an ultra-cold freezer or grind it in a buffer such as Thermo-Fisher Scientific's RNA-Later®, as RNA will degrade rapidly following death. (We note, however, that material preserved in 95–100% ethanol and stored at −80 °C has also yielded satisfactory results for RNA-Seq; *Bazinet et al.*, *2013*).

If the goal is to obtain gDNA, individual specimens can be preserved in a more cost-effective manner by placing them into tubes containing ≥95% ethanol and storing them in a −80 °C freezer. At UMD and MGCL, specimens are stored individually in 2 mL microcentrifuge tubes, 15 mL centrifuge tubes, or 50 mL centrifuge tubes, depending on the specimen's wingspan and body size. Multiple specimens from the same location can be stored together in a single Nasco® Whirl-Pak™ bag containing ≥95% ethanol and a locality label (Fig. 1A). For each Whirl-Pak, the number of specimens is usually kept to ≤10, in order to ensure that the ethanol does not become too diluted. We typically use the 18 oz. (532 ml) capacity 4–1/2″ W × 9″ L (11.5 cm × 23 cm) Whirl-Pak bag, but other sizes can be used.

The use of a Whirl-Pak for mass storage of specimens is not unique to this research. The technique is often applied to large-scale insect diversity studies. Keeping specimens in Whirl-Paks in dense numbers does not significantly harm specimens, as the number of specimens can prevent movement of samples inside the bag. We change the ethanol in the bag while in the field to prevent ethanol dilution, and after several thousand PCRs, we have not experienced any cross-specimen DNA contamination with this procedure. Some scales can occasionally rub off in ethanol and in the transfer process, (note the detached red scales in the clear window, Fig. 1J). Although the procedure cannot prevent some damage, we find that our approach provides the best compromise between large-scale biodiversity sampling, a morphological voucher, and tissues usable for next-gen DNA sequencing. For best DNA quality and quantity results, specimens stored in ethanol should be deposited in a freezer with a temperature below −80 °C as soon as possible.

Alternatively, Lepidoptera specimens can be collected, papered in a glassine envelope, and dried. While there are numerous methods for drying Lepidoptera (e.g., placing specimens in a drying chamber), a simple method is to place the glassine envelope containing the Lepidoptera specimen within a small, airtight, Ziploc® sandwich bag containing silica gel beads, which act as a desiccant. For best DNA recovery using this approach specimens should be collected fresh and placed in the envelope before or very shortly after death, as humidity after specimen death can lead to nucleic acid degradation. At MGCL, we use a 5:1 mixture of clear and colored silica gel beads; the latter change color when saturated with water, thus serving as an indicator that helps determine when the gel needs to be replaced. Two tablespoons of the silica gel mixture is added to each Ziploc bag, placing no more than 20 lepidopterans in a single bag. Silica gel is a great resource because it can be heated and dried for multiple uses.

## Step 2: labeling

Each specimen is given a unique alphanumeric accession code (e.g., "LEP-12345", Fig. 1E) that associates its tissue sample with its corresponding wing voucher. The code is also
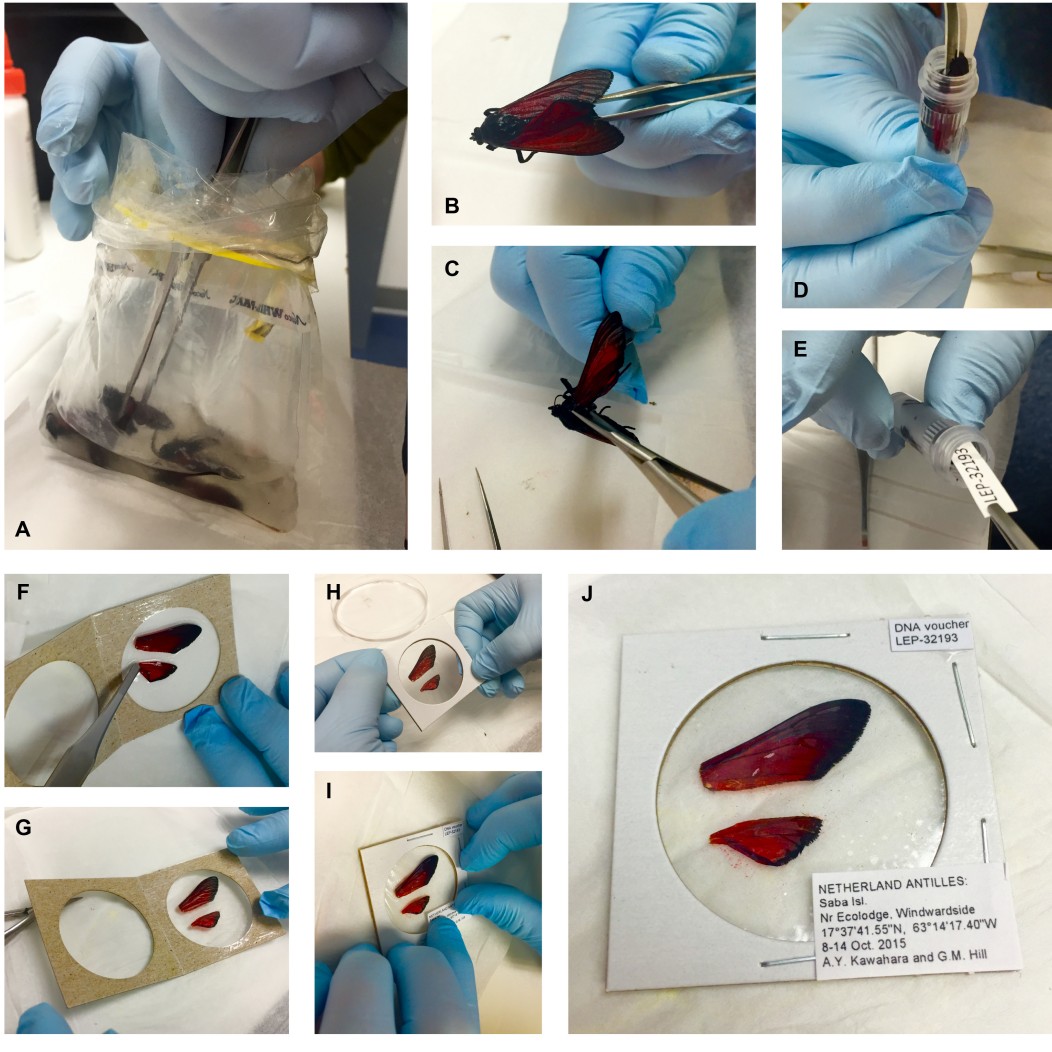

**Figure 1** **Preparation of small to medium-sized Lepidoptera wing vouchers (A–J).** (A) specimen is removed from a Whirl-Pak; (B & C) right wings are cut off at the base and removed; (D) specimen is placed into a storage tube with ≥95% ethanol for DNA preservation; (E) labels are added to the tube; (F & G) wings are placed in the clear window of a coin holder, spread out with forceps; (H) coin holder is closed and edges are stapled to keep the wings secure; (I & J) labels are glued on the top right and bottom right corners of the coin holder.

entered into a collections database and printed on acid-free cardstock with 11-point bold Arial font. Using the same acid-free cardstock, a locality label is printed in 4 to 5 point Arial font. Copies of the code and locality labels are inserted into the specimen tube, and a duplicate copy of each label is saved so that it can be attached to the corresponding wing voucher (see Step 5, below). If specimens were initially collected in a Whirl-Pak, they are gently rinsed with ≥95% ethanol after removal from the Whirl-Pak bag and subsequently transferred to individual tubes containing ≥95% ethanol (see Step 3, below). Ethanol is stored in a −20 °C freezer prior to application in order to prevent warming when the tissues are transferred.

## Wing removal

This step requires a sterile surface, such as a sterile plastic petri dish. The dish, sterilized dissecting scissors and surgical forceps should be prepared before specimens are removed from the freezer, in order to minimize the specimens' exposure to room temperature. When ready for wing removal, a specimen is transferred from its container and placed on the sterile surface. Scissors are used to cut the right forewing and hindwing at the wing base, and both wings are gently removed with forceps (Figs. 1B and 1C). The left pair of wings is not removed; it serves as a secondary identification measure, should the first pair become misplaced. For consistency, we use the right wings as vouchers, unless severe damage to the right wings renders them useless for identification, in which case the left wings are removed and vouchered instead. The remainder of the body is returned to its tube, or placed in a new tube if it was originally stored in a Whirl-Pak (Figs. 1D and 1E).

## Specimen storage

Specimen tubes containing tissue, ethanol, and labels (Fig. 1D) are marked on the tube lid with an appropriate accession code. This process is carried out on a block of ice to keep specimens cold. Fisherbrand® marking pens are used to mark the tubes because their ink is resistant to ethanol "bleeding". Labeled tubes are transferred to a VWR® cardboard freezer box where they are stored upright to prevent leakage, and subsequently returned to the $-80\ °C$ freezer. Microcentrifuge tubes are stored in $5.25'' \times 5.25'' \times 3.00''$ boxes (13.34 cm $\times$ 13.34 cm $\times$ 7.62 cm), whereas taller centrifuge tubes are stored in $5.25'' \times 5.25'' \times 4.87''$ boxes (13.34 cm $\times$ 13.34 cm $\times$ 12.37 cm). Boxes are labeled with the corresponding range of accession codes (e.g., "LEP-12345–12426") on the upper left corner and side of the lid. Two-inch (5.08 cm) cardboard freezer boxes can be used to store 15 and 50 mL microcentrifuge tubes laterally, but this approach will risk the possibility of ethanol leakage.

## Wing vouchering

The wing vouchering step uses two different protocols, depending on the size of the insect. Small specimens with forewing length (FWL) below ~35 mm (Fig. 1) are placed in cardboard coin holders. Large Lepidoptera with FWL >35 mm that do not fit in coin holders are stored in double-laminated polypropylene bags (Fig. 2). Both approaches preserve the wings and allow for easy viewing of both the dorsal and ventral surfaces without causing additional damage, and provide the opportunity for wings to be scanned and digitized. Wings can be removed from the holders if direct examination is required. The remainder of the body is kept in a cryogenic freezer (below $-80\ °C$) for long-term nucleotide preservation. The wing voucher protocol for small and large wings is presented below.

### Small specimens: coin holders

For small specimens, wings are transferred from the sterile petri dish to a clean Kimberly-Clark®, Kimwipe™ in order to remove excess ethanol. Fine dissecting forceps are used to carefully spread each wing, removing any folds or creases, so that the entirety of each wing lies flat against the Kimwipe and is fully visible after it has dried. Wings should be handled by gripping the base, or if necessary, the costa; other regions are more fragile and subject

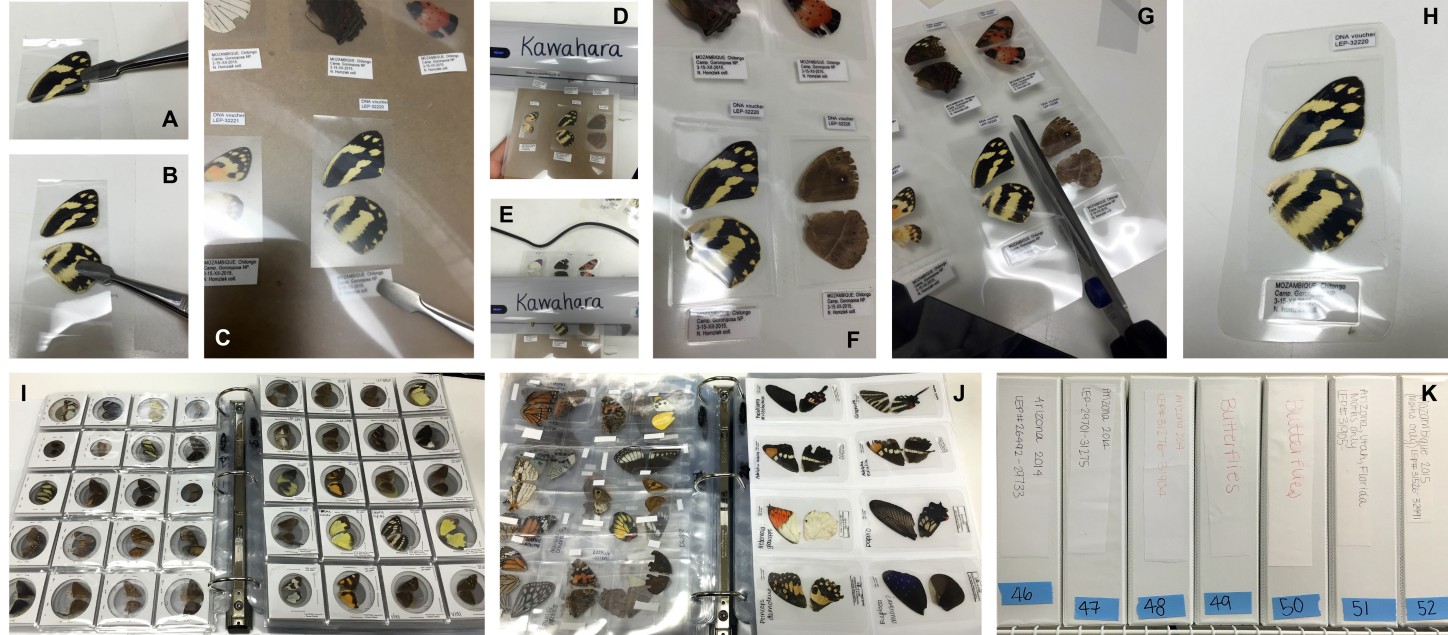

**Figure 2** **Preparation of large Lepidoptera wing vouchers (A–H).** (A & B) following removal of the specimen from ethanol, dried wings are placed inside a BOPP bag, which is then trimmed to wing size; (C) trimmed BOPP bag with wings and corresponding labels is placed within a thermal laminating pouch; (D & E) pouch is laminated 2–3 times to ensure a tight seal; (F–H) voucher is trimmed to size with scissors, cutting around the specimen and labels in a rectangle with slightly rounded edges; (I–K) Stored wing vouchers in binders. (I) Coin holder wing vouchers for small moths; (J) laminated wing vouchers for large moths. (K) Wing vouchers stored in binders on shelf.

to tearing or scale loss. Wings should be dried before transferring them to coin holders (see below), as the clear plastic on coin holders can chemically react and cloud when in direct contact with ethanol. Dried wings are then placed in the clear window of a cardboard coin holder (Figs. 1F–1J). We use BCW®2″ × 2″ (5.08 cm × 5.08 cm) "coin flips", which can be ordered in three different coin sizes: small ("Quarter"), medium ("Dollar"), and large ("Large Dollar"). In order to reduce wing movement within the coin holder, we recommend using the smallest possible window size that will accommodate a given pair of wings. The coin holder is slowly closed and the edges are neatly stapled to keep the wings secure (Fig. 1H). Copies of the corresponding accession code label and locality label (from Step 2) are glued on the top right and bottom right corners of the coin holder (Figs. 1I and 1J). If taxonomic information is known, this is marked in the upper left corner and simultaneously entered in the collections database.

### Large specimens: Lamination

Large Lepidoptera also have their wings dried on a Kimwipe after separation from the rest of the body. Dried wings are placed inside a 4″ × 6″ (10.16 cm × 15.24 cm) Biaxially Oriented Polypropylene (BOPP) bag. The bag is then trimmed by hand to fit the dimensions of the wing (Figs. 2A and 2B). The trimmed bag with the wings is placed within a plastic Scotch® 5″ × 7″ (12.7 cm × 17.78 cm) thermal laminating pouch. The BOPP bag creates a protective layer for the wings and prevents the pouch plastic from adhering to the wing surfaces, enabling wings to be removed from the pouch without damaging them or their

scales. Locality and accession code labels are also placed in the pouch, at the lower left and upper right corners of the BOPP bag, respectively (Fig. 2C). The pouch is then carefully inserted into a Scotch laminator (dial set at 5 mL) and the lamination process is repeated 2–3 times to ensure a tight seal (Figs. 2D and 2E). Each voucher is then neatly trimmed to size, leaving the specimen and labels in a rectangle with rounded corners (Figs. 2F–2H).

### Voucher storage

Vouchers are organized in numerical order using their unique accession code labels, placed into clear pocket pages (ranging from 4–20 pockets per page, depending on voucher size), and stored in 2″ (5.08 cm) three-ring binders with a slanted binder ring (Figs. 2I–2K). Binders are uniquely numbered and labeled with their corresponding range of accession codes and sampling localities. Binder numbers are entered into the collections database so that each sample in the collection can be queried for the location of its wing voucher.

## RESULTS & DISCUSSION

The importance of molecular voucher specimens in modern evolutionary research cannot be overstated. Natural history collections aim to provide a record of Earth's biodiversity and serve as "windows on evolutionary processes" (*Holmes et al.*, *2016*). The preservation of specimens for both morphological and molecular research should be considered a fundamental aspect of the mission of natural history collections.

Unfortunately, a standard for molecular vouchering is still lacking to this day for most museums. The procedure presented herein provides an effective long-term preservation method for Lepidoptera tissues while simultaneously creating easily accessible, compact morphological vouchers for long-term storage and identification. Putative informative characters are preserved as best as possible, and vouchered wings can be removed from its holder, if necessary. Our protocol incorporates one pair of wings; in the event that a voucher is lost, the other pair remains preserved with the remainder of the specimen in the freezer, thereby allowing tissues to be stored permanently with verifiable morphology. It is important to note that when preserved this way, wing color and pattern will be retained, since most wing scales do not rub off in ethanol or during preparation, allowing specimens to used for both traditional and modern taxonomic and phylogenetic studies (genitalia can be easily dissected from ethanol-stored specimens using standard methods including 10% KOH).

Our proposed method for storage of molecular samples is not meant to replace traditional Lepidoptera preservation methods (pinning, spreading wings, and drying), nor is our approach meant to replace the classic Lepidoptera pinned collection. Instead, it is meant to complement traditional approaches and create a measure for long-term, secure DNA preservation. We have examined the potential for DNA degradation between recently dried specimens (as described above) and a tissue taken and placed into $\geq$95% ethanol, and there is no significant difference, with both approaches providing the quality and quantity required for high-throughput sequencing. We propose that our approach be used, for instance, when multiple specimens of the same species are encountered in the field, allowing for specimens to be preserved using both methods. We also realize

that our approach is not applicable to all natural history collections worldwide; there are clear disparities in physical and financial resources available for collection preservation in different countries. Securing biodiversity funding resources to establish an international cooperative repository for genomic quality tissues (e.g., seed banks) would serve as an important step to secure tissues for long-term storage.

One limitation to our approach is its applicability to very small moths. Although we have had success vouchering small moths (e.g., Tortricidae and Gelechioidea), most Microlepidoptera have extremely fragile wings that require delicate care and intricate handling. For very small moths, we suggest that the moth is placed in a small, shallow dish containing ≥ 95% ethanol, and cut the wings of the moth from the body while submerged. After separating the forewing and hind wing from the body and opening their wings in ethanol, excess ethanol is gently dabbed with a Kimwipe and the wings left to dry before transfer to the coin holder. Alternatively, the protocol of *Lopez-Vaamonde et al.* (*2012*) can be used, which entails pinning the moth while fresh, immediately placing the abdomen and legs in a small tube with ethanol, and storing the tube in a cryogenic freezer. Obviously each natural history collection will have their own way of documenting, labeling, and categorizing specimens. Our details and methodology, outlined above, are not "hard and fast" rules, but rather suggestions for which products and approaches have been successfully applied by the authors.

We hope that the procedure presented here will provide a standardized methodology for museum collections that seek to build a Lepidoptera molecular collection. Our vouchering system is flexible and not restricted to any particular database format. It can be used with different management databases such as Specify (http://specifysoftware.org/), Symbiota (http://symbiota.org/docs/), and VoSeq (*Peña & Malm*, *2012*). With the move to digitize natural history collections gathering momentum around the world (*Holmes et al.*, *2016*), we feel this method provides Lepidoptera natural history collections the opportunity to easily distribute specimen data and images on a global scale, while accurately linking specimens to tissue samples.

## ACKNOWLEDGEMENTS

The authors thank the dedicated volunteers, students, staff and visiting scholars who have put in countless hours to build the molecular collections at the University of Maryland and the University of Florida.

### Funding

This paper was funded in part by National Science Foundation grant numbers DBI-1349345, DEB-0531639, DEB-1354585, DEB-1541500, DEB-1557007, and IOS-1121739. The funders had no role in study design, data collection and analysis, decision to publish, or preparation of the manuscript.

## Grant Disclosures

The following grant information was disclosed by the authors:
National Science Foundation: DBI-1349345, DEB-0531639, DEB-1354585, DEB-1541500, DEB-1557007 and IOS-1121739.

## Competing Interests

The authors declare there are no competing interests.

## Author Contributions

- Soowon Cho, Kim Mitter and Charles Mitter conceived and designed the experiments, contributed reagents/materials/analysis tools, wrote the paper, reviewed drafts of the paper, planning and execution of methodology.
- Samantha W. Epstein and Akito Y. Kawahara conceived and designed the experiments, contributed reagents/materials/analysis tools, wrote the paper, prepared figures and/or tables, reviewed drafts of the paper, planning and execution of methodology.
- Chris A. Hamilton wrote the paper, prepared figures and/or tables, reviewed drafts of the paper, planning and execution of methodology.
- David Plotkin wrote the paper, reviewed drafts of the paper, planning and execution of methodology.

## Data Availability

The research in this article did not generate any raw data.

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
