# Peer review of "Preserving and vouchering butterflies and moths for large-scale museum-based molecular research"

_PeerJ, doi:10.7717/peerj.2160_

## Round 0.1 · original submission · Major Revisions

Dear authors,

As you can see two of our reviewers are very critical with your article. The reviewers come from a major museum and work on phylogeny. Thus the concerns are serious. If you are willing to respond to the suggestions and queries we are happy to review your revision.

Kind regards

Michael Wink
Academic editor

·

Basic reporting

No comments

Experimental design

No comments

Validity of the findings

No comments, it is not an experimental work

Additional comments

This paper is dealing with a normalization (or at least a strong suggestion) of how to keep Lepidoptera specimens in Museums. The goal is to provide a technical base in order to conserve both morphological and DNA possibilities to characterize individuals.
This article is clear and well written but as there are no scientific experimental results, it is of course rather difficult to evaluate such work.
With the necessity to conserve suitable tissues to extract DNA, all our ancient conception of museology have to be changed. Samples are not anymore only there to take part of a collection but also to be used for further genetic studies.
The difficulty is to reach an equilibrium between the primary mission of Museums and new constraints.
In this direction, norms established and described in this work seem efficient and reasonable. Even if the main risk in such split of biological sample parts is of course the possibility to mix information.
Authors thank all the volunteers who were performing all this work (which is totally fair) but by multiplying staff, I wander if the risk of mixing samples is not too high. As it is a crucial part of this work, I think that a special section concerning this point have to be developed in your writing (even if parts of the text are already dealing with this problem).
Another point which is not at all developed in this work is the feasibility of this standardization. All the procedure to conserve DNA samples is costly and I’m afraid that only some museums (and even not all) of first world countries will be able to perform all these procedures (considering technical help = salaries, vials, coin holders…..and…much more costly cryogenic freezers). In my mind, we may imagine that no more than 5 museums in all Latino America and 1 or 2 in Africa will be able to apply this standardization.
I think that authors have to propose/explore in this paper solutions (at the international level) in order to apply this standardization which in my mind is a good and necessary measure. What about international banks such as those already existing for seeds? Local museums will keep albums with wings while sensible DNA material will be stocked in such “safe” places financed by international funding?
Of course the risk can be what was happening in Alep/Syria where it seems that most of samples of seeds of Mediterranean plants are lost.
I think that the proposal described in this work is very interesting and that all of us working on Lepidoptera have to be aware of such technical suggestions. It is why I recommend strongly the publication of this work. Anyway, in two, in my mind, crucial points, I think that authors have to be more direct and clear to secure the feasibility of this museum necessary evolution/revolution.

Reviewer 2 ·

Basic reporting

Review of Cho et al., submitted to Peer J
A procedure for preserving and vouchering butterflies and moths for large-scale museum-based molecular research (#9427)

The authors describe their procedure of preserving Lepidoptera for molecular studies. The methodology appears to have been widely used in their research. However, it does not describe original primary research: no hypotheses are formulated or tested. Research data could theoretically be added, if differences in DNA-content of different methods would be compared or amounts of scale loss with different handling procedures were reported.

With regard to content, the reference to „museum-based“ research should be ommited, since the approach is not qualified for this aspect:
The procedure appears unsuitable for morphological studies, which form the foundation of museum-based research. Scales of wings rub off in ethanol and direct contact to plastic covers. Museum-based researches have this experience. The authors appear to lack this.
The authors adopt a purely technical and mechanistic approach with the goal of molecular analyses, showing lack of finesse with respect to museum- and organism-based research.

I recommend rejection of the manuscript, since it does not report research results.

Resubmission should be considered if research data are added and one aspect of the methodology is revised radically: drying Lepidoptera alive is neither suitable for field trips nor possible for large species with strong muscles that would move destructing morphological characters during their death agony. Apart from these arguments the approach cannot be advocated from an ethical point of view.

Experimental design

There is none. There is also no research question.

Validity of the findings

There are none.

Additional comments

Specific comments

Line 107
This is not suitable for field requirements and also hardly possible for strong bodied species (as hawkmoths). Apart from that it is also rather gruesome and thus I do not recommend publication of this method.
Have you tested the amount of DNA-degradation if you shock-freeze the specimens to kill them?

Line 97
How do you prevent mixing of tissues and leaking body fluids?
How do you prevent scales from being wiped of the specimens by the others in the bag?

Line 164
Are these windows sticky - would the scales get stuck to them? If not, does it happen due to electronic charging of plastic materials? Both of these would destroy the wing scaling pattern, possibly an important source of morphological characters.

·

Basic reporting

No comments

Experimental design

This is a technical report and the details given are sufficient to follow. I think that the authors should use metres as a measure, here cm. Furthermore, I do not think that details on the materials (i.e. catalogue numbers) are necessary.

Validity of the findings

Not applicable here.

Additional comments

I think that the idea presented here as an optimisation for collecting vouchers for morphology and genetics is somewhat dangerous. I do not think that this way of sampling is an equivalent for the classical Lepidoptera collection. Furthermore, this collection is completely depending on freezers, i.e. ob technology and hence much more vulnerable than a classical insect collection. Therefore, I think the authors should elaborate considerably more the shortcomings of their idea. Such a sampling might be acceptable if the primary interested is the genetic material, a bit as an add on the option only preserving the material for genetic analyses. However, I cannot agree with it as a kind of substitute for the classical Lepidoptera collection, which is much more than what is presented by the authors in their "Lepidoptera collection". Elaborating this in details for me is essential for making this technical report acceptable for publication.

---

## Round 0.2 · Minor Revisions

Dear authors

Thank you for resubmitting your manuscript to our journal. As you see our reviewers suggest some remaining minor revisions of your ms. If you are willing to do so, we would be happy to reconsider your revised manuscript.

Michael Wink

Reviewer 2 ·

Basic reporting

The authors provided a revised copy of their manuscript and take into account most of the concerns voiced in the first round of the review process. I acknowledge that the authors made clear from the beginning that the ms contains no original research, but the journal does not allow for pure methods-papers yet. I adhered to answering the questions posed for the review process. Nevertheless, I encourage addition of this category to the scope of the journal.

Experimental design

Not applicable.
I encourage addition of a category "New methods" to the scope of the journal.

Validity of the findings

Not applicable.
I encourage addition of a category "New methods" to the scope of the journal.

Additional comments

Whereas the authors admit (in the original version of the manuscript and in the response to reviews) that “We cannot prevent some damage from occurring”, they removed this modesty in the revised manuscript. However, admitting the risks of their method would greatly increase the impression that they maturely aim at fulfilling the mission of natural history museums, which is still based on morphological studies.
The authors claim to have experience in morphological work. They appear not to have experienced the case of having to study scales of a certain shape in an ethanol preserved moth, of which exactly all of the scales of this shape were rubbed off. The authors’ comments on my question about the risk of cross-contamination show that, contrary to their claim in the revised version of the text, they acknowledge that scales do rub off in the ethanol, which needs to be included in the text.
I made some detailed suggestions below that would allow acceptance of the manuscript for publication.
Line 34
Please change “can be easily viewed without damage” to “can be easily viewed without further damage”
Line 115
Please change “Our approach provides the best compromise between large-scale biodiversity sampling, a morphological voucher, and tissues usable for next-gen DNA sequencing.”
to “Some scales are nevertheless initially lost in the ethanol, additional ones during the drying process and the placement in the protective sleeves. Although the procedure cannot prevent some damage, we find that our approach provides the best compromise between large-scale biodiversity sampling, a morphological voucher, and tissues usable for next-gen DNA sequencing.”
Line 131
I would welcome if you added a sentence here, citing the “other published studies”, to compare the two methods described as stated in the response letter: We have examined DNA degradation between a dried specimen and a tissue taken and placed into ≥95% ethanol and there is no difference. These results are in line with other prior, published studies.
Line 173
Please change “Both approaches preserve wings without causing additional damage, allow for easy viewing of both the dorsal and ventral surfaces, and provide the opportunity for wings to be scanned and digitized.”
to “Both approaches preserve wings and allow for easy viewing of both the dorsal and ventral surfaces without causing additional damage, and provide the opportunity for wings to be scanned and digitized.”
Line 227
Please change “identification – preserving putative informative characters and allowing for vouchered wings to be removed if necessary.”
To “identification. Putative informative characters are preserved as good as possible, although some loss of scales is inevitable. Vouchered wings can be removed if necessary.”
Line 231
The authors themselves falsify this statement in their response: “There are no contamination issues with this procedure as scales do not contain DNA.”.
Please change “It is important to note that when preserved this way, wing scales do not rub off in ethanol and specimens can be used for both traditional and modern taxonomic and phylogenetic studies”
to “It is important to note that when preserved this way, wing color and pattern will be retained, since most wing scales do not rub off in the ethanol. Although a few additional scales also become detached during the process of placing the wings in the coin holder (note the detached red scales in the clear window, Fig. 1J), specimens can be used for both traditional and modern taxonomic and phylogenetic studies”
Line 237
Please change “pinning, drying, and spreading wings” to “pinning, spreading wings, and drying”

·

Basic reporting

I have no comments in the Basic Reportings

Experimental design

Not applicable here

Validity of the findings

Not applicable here

Additional comments

This new version of the manuscript is advance since I have read the first version. I have to state that I am highly critical about this method and therefore, I think that it was highly important to emphasize that the technique presented here cannot substitute the classical butterfly collection. I think this could be emphasized a bit stronger.

Furthermore, I think that many unnecessary details are still given. Thus, it is not relevant in which fond the lables were printed by the authors of this contribution. Everyone will do it in a different style. Further, nobody will be interested about the size of the boxes stored in the freezer at the end. Every group will use something and also something different. Etc etc ... These unnecessary pieces of information should be deleted.

Additionally, I disagree strongly that materials should be given with catalog numbers. In many cases, each scientist will use something available. I think that detailed information on the source used by the authors only should be given if this information is really necessary. In case of materials that can be ordered from many suppliers, I think it must not be mentioned. For me, the way of presentation in this contribution is like writing down the catalog number of eppendorf tubes .... Nobody is e.g. interested that Kimberly-Clark (r), KimwipeTM (Cat. #34155) were used to remove excess of ethanol ....

L 237-240: moths (FWL < 5 mm) are not "very small", they are just small.

---

## Round 0.3 · accepted · Accept

Dear authors

Good news- your revision is accepted and your paper will be published soon. Thanks for submitting your work to our PeerJ.

Regards

Michael Wink
Academic Editor